# Botulinum Toxin A as an Adjunct for the Repair Giant Inguinal Hernias: Case Reports and a Review of the Literature

**DOI:** 10.3390/jcm13071879

**Published:** 2024-03-25

**Authors:** Sergio Huerta, Roma Raj, Jonathan Chang

**Affiliations:** 1Department of Surgery, VA North Texas Health Care System, University of Texas Southwestern Medical Center, Dallas, TX 75390, USA; roma.raj@utsouthwestern.edu; 2Department of Anesthesia and Pain Management, VA North Texas Health Care System, University of Texas Southwestern Medical Center, Dallas, TX 75390, USA; jonathan.chang@va.gov

**Keywords:** Botox, phrenectomy, component separation, preoperative progressive pneumoperitoneum, femoral hernia

## Abstract

The management of giant inguinoscrotal hernias remains a challenge as a result of the loss of the intra-abdominal domain from long-standing hernia contents within the scrotum. Multiple techniques have been described for abdominal wall relaxation and augmentation to allow the safe return of viscera from the scrotum to the intraperitoneal cavity without adversely affecting cardiorespiratory physiology. Preoperative progressive pneumoperitoneum, phrenectomy, and component separation are but a few common techniques previously described as adjuncts to the management of these massively large hernias. However, these strategies require an additional invasive stage, and reproducibility remains challenging. Botulinum toxin A (BTA) has been successfully used for the management of complex ventral hernias. Its use for these hernias has shown reproducibility and a low side effect profile. In the present report, we describe our institutional experience with BTA for giant inguinal hernias in two patients and present a review of the literature. In one case, a 77-year-old man with a substantial cardiac history presented with a giant left inguinal hernia that was interfering with his activities of daily living. He had BTA six weeks prior to inguinal hernia repair. Repair was performed via an inguinal incision with a favorable return of the viscera into the peritoneum. He was discharged on the same day of the operation. A second patient, 78 years of age, had a giant right inguinoscrotal hernia. He had a significant cardiac history and was treated with BTA six weeks prior to inguinal hernia repair via a groin incision. Neither patient had complaints nor recurrence at 7- and 3-month follow-ups. While the literature on this topic is scarce, we found 13 cases of inguinal hernias treated with BTA as an adjunct. BTA might be a promising adjunct for the management of giant inguinoscrotal hernias in addition to or in place of current strategies.

## 1. Introduction

One of the most common operations performed by general surgeons globally is inguinal hernia repair. This operation alone accounts for over 20 million surgeries performed annually worldwide [1]. The management of inguinal hernias has rapidly evolved over the past two decades to include the following: (1) watchful waiting for small asymptomatic hernias [2]; (2) open repair (with or without mesh, under local or general anesthesia) [3]; and (3) laparoscopic repair [total extraperitoneal repair (TEP), transabdominal peritoneal repair (TAPP)] [1] and robotic repair [4]. However, the management of giant inguinal hernias requires special attention. Giant inguinal hernias are defined as hernias with an inguinoscrotal component extending to midthigh in the standing position [5,6]. Years of neglect and poor follow-up are common for these patients. Consequently, comorbid conditions often complicate the management of these individuals.

These hernias significantly alter the quality of life of patients. The massive inguinoscrotal component might lead to (1) scrotal pressure ulcers [6,7]; (2) voiding difficulties and urinary retention [8]; (3) difficulties with urination as a result of burrowing of the penis within the inguinoscrotal component [7]; (4) problems with walking, sitting, and lying down [6,9,10]; and (5) the typical issues associated with hernias, which include incarceration and strangulation [7] as well as bowel obstruction and even perforated viscus within the hernia sac [1]. Contents within the hernia sac might also pose a challenge in the management of these hernias. Just about any intra-abdominal structure can find its way within the hernia sac [11].

The return of abdominal contents in a loss-of-domain cavity might result in a sharp increase in intra-abdominal pressure, compromising cardiac and respiratory physiology [12]. In extreme cases, this might lead to abdominal compartment syndrome and even compromised venous return from the lower limbs by compression of the inferior vena cava [13].

Thus, multiple strategies to preoperatively relax and enlarge the abdominal cavity have been described, including preoperative progressive pneumoperitoneum (PPP), postoperative ventilation with relaxation, resection of viscera with abdominal wall reconstruction, and other techniques [6,12]. All of these techniques require a second invasive intervention and have their own associated complications and an increased length of hospital stay [14].

Botulinum toxin A (BTA) is a neurotoxin that selectively acts on presynaptic nerve terminals. BTA application results in temporary muscle paralysis without systemic side effects [15]. Application of BTA to the lateral abdominal muscle complex (internal/external obliques and transversus abdominus) relaxes the abdominal wall. It reaches its maximum effect 2–4 weeks following injection, but its effects last up to 6 months [16]. This muscle relaxation achieved by BTA has been used successfully for the management of complex ventral hernia closure [16,17,18].

While BTA has been previously reported as an adjunct for the management of inguinal hernias, the literature on this topic remains scarce [13]. In the present report, we discuss our institutional experience with the management of giant inguinal hernias for which BTA was successfully employed, leading to the repair of these hernias via an inguinal incision without complications.

## 2. Materials and Methods

The following reports are of two patients who presented to the Veteran Affairs (VA) North Texas Health Care System with giant inguinal hernias. The case history was reviewed in the Computerized Patient Record System (CPRS) at the institution. Informed consent was obtained from both patients for the publication of these reports, as well as permission to include their photographs in the publication. Both patients underwent management of their inguinal hernias by the same surgeon (SH).

### 2.1. Review of the Literature

A comprehensive review of the literature was undertaken. The initial literature review was performed by SH on 15 January 2024. Various combinations of keywords, including “hernia”, “inguinal hernia”, “giant hernia”, “inguinoscrotal hernia”, “femoral hernia”, “groin”, and “botulinum toxin A [Botox^®^ (Allergan plc, Ivine, CA, USA), Dysport ^®^ (Ipsen Biopharm Limited, Cambridge, MA, USA), Xeomin^®^ (Merz Phamaceuticals, Raleigh, NC, USA)”, were used for our searches. No time restriction (beyond that of the existing databases) or language restriction was imposed. Databases including PubMed, MEDLINE (via PubMed), and Embase were initially queried. Subsequently, Google, Google Scholar, and ResearchGate were utilized to search and acquire reports that were new and/or unavailable from the previous databases. Any hernia where BTA was used was the primary inclusion criterion. Publications where BTA was used for other hernias (i.e., ventral hernias) were excluded from the final analysis. Further manuscripts were identified by close examination of the references of the index papers and main reviews on this subject. These manuscripts were included in our review if they were appropriately referenced and did not duplicate our original findings of patients.

The PRISMA flow chart depicts the screening process (Figure 1). All the abstracts were analyzed with an EndNote group to eliminate irrelevant and duplicated studies. Google Translate was utilized to translate articles in other languages. SH is also fluent in Spanish and read the abstracts and papers in that language to be included in the analysis.

### 2.2. Case Reports

#### 2.2.1. Case 1

A 77-year-old white male veteran with a past medical history significant for hypertension, hyperlipidemia, chronic obstructive pulmonary disease (COPD), congestive heart failure, atrial fibrillation, and an active smoking history for more than 40 years presented to our clinic with complaints of a left inguinal hernia. He had a body mass index (BMI) of 27.4 kg/m^2^. He first reported noticing a bulge in the left inguinal region many years ago that had since increased in size. He denied pain or episodes of acute on chronic incarceration, such as signs consistent with obstructive symptoms (i.e., nausea and vomiting or constipation). On examination, the patient had a large left inguinal hernia reaching to his knees with no overlying skin changes. It was minimally reducible and non-tender (Figure 2).

We discussed the potential benefits of BTA administration. He received BTA injections to the abdominal wall on 13 July 2023 and underwent open left inguinal hernia repair via a groin incision on 4 August 2023. BTA was provided by the Anesthesia and Pain Management service at the VA North Texas Health Care System under a protocol described by Deerenberg et al. [19].

Briefly, 200 units of BTX (Botox^®^) diluted into 100 milliliters (mL) of saline (2 Units/mL) were used. These were divided into 8 mL aliquots in 12 separate 10 mL syringes (16 units of BTX per syringe). The patient was placed in a supine position, and 3 equidistant points were identified in the anterior axillary line of the abdomen between the bottom of the rib cage and the top of the anterior superior iliac crest bilaterally. Then, a high-frequency linear ultrasound transducer (10 MHz or greater) was used to identify both the internal and external oblique muscles at each of the above points. After sterilely prepping the overlying skin with 2% chlorohexidine gluconate and 70% *v*/*v* isopropyl alcohol, a 25-gauge 1.5-inch needle was initially placed into the internal oblique muscle using an out-of-plane technique, and 16 units of BTX were injected after negative aspiration. After this, the needle was withdrawn into the external oblique muscle, and 16 units of BTX were injected after negative aspiration. This procedure was then repeated in a similar fashion at the 5 other points previously identified. No complications were noted with this procedure. The operation was scheduled four weeks after injection.

During the operation, following the incision of the aponeurosis of the external oblique, the cord structures were exposed. Fibers of the cremasteric muscle were divided, and then the hernia sac was identified. It was then extracted from the scrotum and carefully delivered into the groin incision. The sac was not violated. A massive left indirect inguinal hernia with small bowel loops in the sac and a completely obliterated floor was identified. Once all the scrotal contents within the sac were delivered into the groin incision, they were readily reduced into the relaxed abdominal cavity. The floor was recreated using proline mesh, and the rest of the hernia was repaired following a standardized technique by SH, as previously described [3]. There was no need for a laparotomy to accommodate the bowel, and the abdomen was sufficiently relaxed to allow all abdominal contents within. The patient tolerated the procedure well and was transferred to recovery and discharged home the same day. On postoperative follow-up, the patient was doing well with no complaints. His incision was healing well, and he had no postoperative complications. He was doing well at a 6-week follow-up in the clinic. There was no evidence of recurrence or inguinodynia at a 7-month follow-up on a phone call.

#### 2.2.2. Case 2

The patient is a 78-year-old white male veteran with a past medical history significant for hypertension, diabetes mellitus type II, atrial fibrillation (managed with pacemaker as well as apixaban), coronary artery disease, and a history of urinary bladder cancer that was resected 10 years prior. He had a BMI of 20.3 kg/m^2^. He presented to our clinic in January 2023 with a giant right inguinal hernia that had progressively enlarged in size. His symptoms included discomfort and difficulty walking. On examination, he had a right inguinal hernia with a large scrotal component that reached his knees while standing and had no overlying skin changes. It was partially reducible when lying down, and there was non-tenderness (Figure 3).

During the initial visit to our clinic, he deferred surgery due to concerns about complications related to surgery. He subsequently presented to an outside facility in August 2023, where laparoscopic repair of the hernia was attempted but aborted due to the large bowel component in the hernia sac and an inability to reduce the inguinoscrotal contents in the abdominal cavity.

When he re-presented to our clinic, we discussed the potential benefits of preoperative BTA injection administration. He received BTA, as described in Case 1, four weeks prior to inguinal hernia repair.

He underwent an open right inguinal hernia repair with mesh placement via a groin incision on 12 January 2024. Operative findings revealed a right indirect sliding inguinal hernia, with almost all the bowel within the sac and an obliterated floor. The operation was performed in a similar fashion as in Case 1. The deep ring had to be opened substantially to effectively accommodate and reduce the sac contents. The floor was repaired, and the deep ring was recreated with proline mesh. There was no need for a laparotomy to accommodate the bowel, and the abdomen was sufficiently relaxed to allow all abdominal contents within. He tolerated the procedure well and was transferred to recovery and then to the inpatient unit for observation overnight, given his extensive comorbidities. He was doing well at a six-week follow-up in the clinic and a three-month follow-up on a phone call without evidence of inguinodynia or recurrence.

## 3. Discussion

Compromised cardiorespiratory physiology remains a challenge in the management of giant inguinoscrotal hernias. With massive inguinal hernias, this reduction might be impossible without some form of preoperative planning for elective repair [20]. Several techniques have been described for the management of giant inguinoscrotal hernias, which include the following: (1) abdominal wall component separation following reduction [21,22]; (2) maintenance of postoperative intubation with paralysis up to four days following reduction of viscera [20]; (3) PPP [23,24]; (4) resection of abdominal contents (debulking) and use of mesh and scrotal skin flaps for abdominal wall reconstruction [9]; (5) creation of an abdominal wall defect (phrenectomy) and use of the inguinal hernia sac with surgical repair of both the inguinal and the ventral defect [10,12]; (6) laparoscopic component separation [25]; and (7) abdominal wall expansion utilizing tissue expanders [26]. A combination of multiple techniques has also been employed in many cases [5,6,12,21,25,27].

Unfortunately, previously published reports are commonly inclusive of a single case in which the surgeons report their experience. Only two series involving eight patients each have been reported. In one, a novel technique was introduced to create a hernia defect, pulling the sac into the abdominal cavity, fashioning the sac to augment the peritoneum, and closing the hernia defect with mesh [12]. In 2020, Tang et al. described a technique of a combination of PPP and BTA for giant inguinoscrotal hernias [28]. While these techniques showed suitable results in these cohorts of patients, their reproducibility by other surgeons might be difficult to implement. Additionally, all techniques involve a multi-step approach and require a specific set of skills by surgeons.

Thus, alternatives for adjunct modalities are needed. BTA has emerged as an additional strategy as an adjunct in the management of giant inguinal hernias in recent years. This strategy was implemented and extrapolated from previous experience in the management of complex ventral hernias [15]. BTA as an adjunct modality for the management of giant inguinoscrotal hernias has been previously described, but the literature remains scarce [13,28,29,30].

The first case of BTA as an adjuvant for the management of a giant inguinoscrotal hernia was reported by Ibarra-Hurtado et al. in 2014 [17]. In this report, the authors successfully managed a 66-year-old man with giant bilateral inguinal hernias. BTA was applied as a single injection four weeks prior to the operation (Table 1). The patient was discharged 24 h after the operation and was doing well at the 46-week follow-up visit. In this paper, the authors also provided a comprehensive review of all modalities that increase the intra-abdominal domain for the management of complex ventral hernias. The authors conclude that all other modalities carry many more risks compared to the use of BTA for the management of large inguinoscrotal hernias [17].

In four prior reports, BTA was used in combination with PPP [28,29,30,31]. One report described the successful management of giant inguinoscrotal hernias in eight patients with a combination of PPP and BTA. Notably, all these hernias were successfully repaired laparoscopically (via the TAPP approach) [28].

Apart from the initial report describing the exclusive use of BTA for inguinal hernias [17], only a second report was found in the literature using BTA as the only adjunct for the management of a large inguinoscrotal hernia. This case was on a relatively young patient who had a successful outcome [13]. In all, 13 patients were found in the literature to employ BTA as an adjunct for large inguinoscrotal hernias, and only 2 were without any other modality (Table 1).

Adverse outcomes with the use of BTA for the management of ventral hernias have demonstrated a low side effect profile [32]. Minor side effects include a weak cough, back pain, and superficial bruising at the site of the injection [33], all of which are substantially less than previously established techniques to augment the abdominal wall domain [15]. In patients with back issues, omitting the injection into the transversus abdominis has been shown to decrease this side effect [34].

Clear indications for BTA have not been established in the management of complex ventral hernias nor inguinoscrotal hernias [15]. Indications, dosage, timing, and efficacy of BTA for inguinal hernias remain obscure. The present report adds two cases to the world literature. We report two elderly patients (>70 years of age) with substantial comorbid conditions for whom other modalities might have carried additional morbidity. We utilized the technique initially described by Deerenberg et al. [19]. However, the technique originally described by Ibarra-Hurtado et al. has been widely cited [17]. In contrast to the two previously reported cases (Table 1, patients #1 and #12), our patients were older and with more comorbid conditions. One patient was discharged the same day, while the other stayed in the hospital for 24 h. None of these patients experienced any intraoperative or postoperative complications. The use of BTA was well tolerated and did not cause any complications.

Compared to all current modalities to accommodate inguinoscrotal components into a loss-of-domain peritoneal cavity, BTA is the least invasive and most likely to be tolerated. BTA administration is feasible and reproducible and has demonstrated no substantial complications in other similar applications, such as ventral hernias. At this juncture, the limited literature is not sufficient to provide firm recommendations, nor does this report suggest that BTA should replace current techniques available to surgeons in the management of large inguinoscrotal hernias. However, this report, along with two other publications, demonstrates the feasibility of this technique as an adjunct for the management of giant inguinoscrotal hernias.

The use of BTA for ventral hernias further supports BTA injections as a reproducible technique with a low side effect profile. Further studies will need to be undertaken to provide recommendations for the dose, timing, injection sites, and cost-effectiveness of this modality for the management of giant inguinoscrotal hernias.

There are several limitations to the present report. One of the most substantial limitations of our retrospective review emanates from the inability to accurately retrieve all the data from case reports. However, this is a limitation of all reviews of the literature. There was only a limited number of cases in the literature reporting BTA for the management of inguinal hernias (*n* = 13). This low number of cases is probably a reflection of the rare occurrence of large inguinoscrotal hernias in high-income countries (HICs). In HICs, complex ventral hernias are more common than giant inguinal hernias. However, BTA has been successfully used in HICs for ventral hernias. While giant inguinal hernias might be much more common in low- and middle-income countries (LMICs), the cost of BTA might be prohibitive for its applicability in these regions. Thus, publication bias is a highly likely limitation of this study.

## 4. Conclusions

The current evidence for the use of botulinum toxin A as an adjunct for the management of giant inguinoscrotal hernias is substantially limited. However, the experience with the management of ventral hernias and the few reports in inguinal hernias are promising as an adjunct strategy with excellent reproducibility as well as a low side effect profile. Further studies will need to be undertaken to determine rigid guidelines and cost-effectiveness. Our institutional experience showed excellent tolerability in older patients with a burden of comorbid conditions.

## Figures and Tables

**Figure 1 jcm-13-01879-f001:**
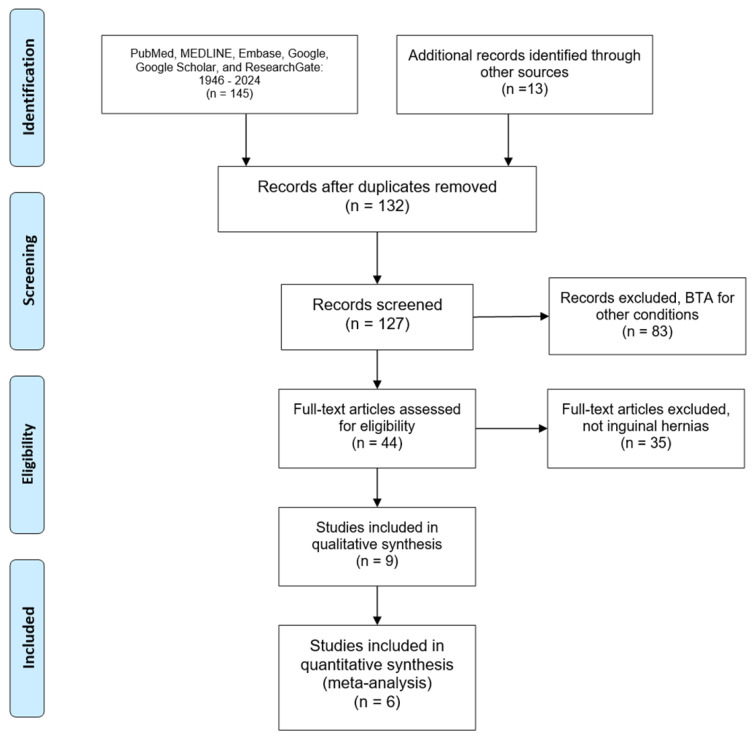
PRIMA flow chart describing the process of our literature search. BTA = botulinum toxin A.

**Figure 2 jcm-13-01879-f002:**
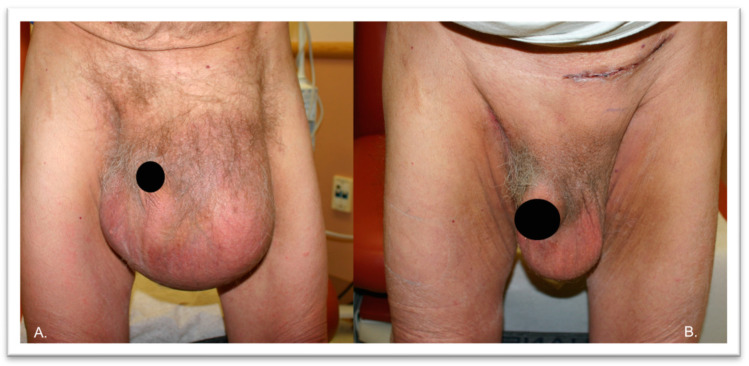
Patient #1 shows a giant left inguinal hernia before (panel (**A**)) and after the operation (panel (**B**)).

**Figure 3 jcm-13-01879-f003:**
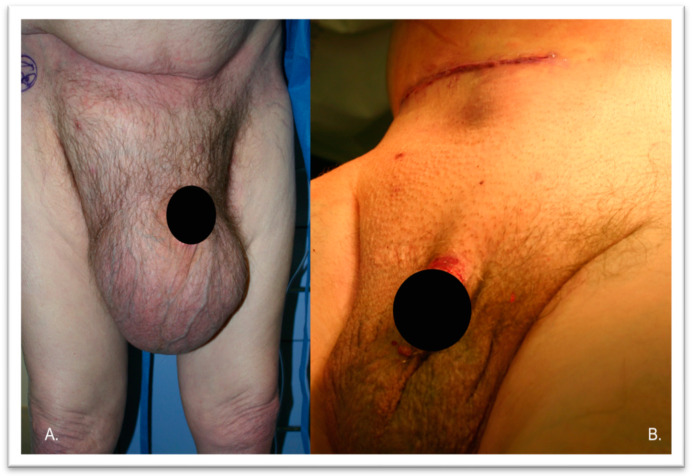
Patient #2 shows a giant right inguinal hernia before (panel (**A**)) and after the operation (panel (**B**)).

**Table 1 jcm-13-01879-t001:** Patients’ characteristics.

Pt. #	Reference	Age	Laterality	Comorbidities	Timing of Botox	BTA Used and Dose	Type of Incision	Laparotomy	Hospital LOS	Complications
1	Ibarra-Hurtado 2014 [17]	66	Bilateral	Smoker	45 d	Dysport^®^ 500 U	Inguinal	No	24 h	None at 46 mo
2	Palmisano 2017 [31] *	76	Right	HTN, HLP, BPH, CVA	21 d	Xeomin^®^ 200 UI	Inguinal	No	12 h	None at 1 mo
3	Gonzalo 2019 [30] *	54	Right	NR	21 d	Dysport^®^ 500 U	Inguinal	No	NR	None at 2 yrs
4–11	Tang 2020 [28] *	58–73	4 R, 2 L, 2 B	NR	2–3 wks	Botox^®^	Laparoscopic (TAPP)	No	7–9 d	Seromas, hematomas
12	Lucas-Gurrero 2020 [13]	57	Left	Smoker, HTN	4 wks	Dysport^®^ 500 U	Midline	Yes		None at 4 wks
13	Menenakos 2020 [29] *	65	Bilateral	Developmental delay	NR	Botox^®^ 100 U	Stoppa	Yes		None at 15 mo
14	Huerta 2024 ^#^	77	Left	HTN, HLP, COPD, CHF, A-fib, smoker	4 wks	Botox^®^ 200 U	Inguinal	No	24 h	None
15	Huerta 2024 ^#^	78	Right	HTN, DM, A-fib, CAD	4 wks	Botox^®^ 200 U	Inguinal	No	None	None

BTA = botulinum toxin A, TAPP = transabdominal peritoneal approach, R = right, L = left, B = bilateral, U = units, HTN = hypertension, HLP = hyperlipidemia, BPH = benign prostate hyperplasia, CVA = cerebrovascular accident, h = hours, d = days, wks = weeks, mo = months, yrs = years, NR = not reported, COPD = chronic obstructive pulmonary disease, CHF = congestive heart failure, LOS = length of stay, DM = diabetes mellitus. CAD = coronary artery disease. * These patients were treated with a combination of preoperative progressive pneumoperitoneum and botulinum toxin A. ^#^ Current cases.

## Data Availability

No new data were created or analyzed in this study. Data sharing is not applicable to this article.

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
