# Peer review of "Botulinum Toxin A as an Adjunct for the Repair Giant Inguinal Hernias: Case Reports and a Review of the Literature"

_jcm, 2024, doi:10.3390/jcm13071879_

Round 1

Reviewer 1 Report

Comments and Suggestions for Authors

In this report, the authors describe their experience with the treatment of two patients with giant inguinal hernias in whom BTA was used successfully and led to complication-free repair of these hernias via an inguinal incision. The authors also reviewed the literature.

I read the study with great interest. The paper is well designed and written, congratulations to the authors. The topic is relatively new and of interest. The literature on this topic is still scarce. The most valuable part of this paper is the literature review. I have several suggestions for improvement.

1. Abstract – The abstract is mostly general and does not relate to the cases at hand. The authors should write a few short sentences about the cases presented (age of patients, symptoms, treatment modalities)

2. Methodology – Authors should state the exact date on which they undertook a comprehensive review of the literature, together with the initials of the authors involved.

3. Methodology – Authors must define precise inclusion and exclusion criteria for the literature search. In addition, they should provide a flowchart clearly stating the number of papers identified in each database, the number of duplicates and the number of papers excluded along with the reasons for exclusion and finally the number of studies included.

4. Case reports – Please provide the exact follow-up period for each patient. The authors state that both patients were doing well at the 6-week follow-up visits. Since the surgery was performed in 2023, I would advise you to define the maximum follow-up time for each patient more precisely. This is important for further literature reviews or meta-analyses.

5. Abbreviations should be expanded when they are mentioned for the first time in the text (e.g. VA, COPD, BMI, etc.).

6. Abbreviations that have been explained once in the text, such as preoperative progressive pneumoperitoneum (PPP), should not be explained again in the Discussion. In the Discussion, authors should only use abbreviations, otherwise the use of abbreviations makes no sense.

7. Table 1 – Each abbreviation used in Table 1 should be presented in a table legend. Some abbreviations have not been explained (e.g. BTA, LOS, CAD, DM, etc.). The legend should be placed below the table, not above it.

8. The authors should add a few lines in the discussion of how botulinum toxin is helpful in treating these giant hernias and the benefits of this treatment compared to treatment without the use of botulinum toxin.

Author Response

In this report, the authors describe their experience with the treatment of two patients with giant inguinal hernias in whom BTA was used successfully and led to complication-free repair of these hernias via an inguinal incision. The authors also reviewed the literature.

Response: We thank the reviewers for the diligent and thoughtful comments regarding our manuscript.  We appreciate the opportunity to continue a discussion on this important subject regarding the limited literature on the use of Botulinum toxin A (BTA) for the management of giant inguinal hernias.   The time and effort that the reviewers have placed on this work is highly evident and a good testament of excellent peer-reviewing practices.  Being a reviewer myself, I appreciate the feedback immensely.  We will attempt to be as diligent with the answers and the revision of our manuscript to accommodate all the concerns from the reviewers and simultaneously add clarity and meaning to this manuscript.   

At the heart of this discussion is the limited information available for a new technique.  Importantly, we have found that BTA for complex ventral hernias is a common current practice in high-income countries (HICs).  However, large inguinoscrotal hernias are less common in HICs compared to ventral hernias.  In low- and middle- income countries, giant hernias are more common, but the availability of BTA is less accessible, probably because of cost.   We hope to address all these issues along with the comments from the reviewers in the revised version of our manuscript.   

I read the study with great interest. The paper is well designed and written, congratulations to the authors. The topic is relatively new and of interest. The literature on this topic is still scarce. The most valuable part of this paper is the literature review. I have several suggestions for improvement.

Response: Thank you. 

  1. Abstract – The abstract is mostly general and does not relate to the cases at hand. The authors should write a few short sentences about the cases presented (age of patients, symptoms, treatment modalities)

RESPONSE:  This has been added to the abstract accordingly.   We added some information on the literature review as well.   

  1. Methodology – Authors should state the exact date on which they undertook a comprehensive review of the literature, together with the initials of the authors involved.

RESPONSE:  This has been added accordingly.  

  1. Methodology – Authors must define precise inclusion and exclusion criteria for the literature search. In addition, they should provide a flowchart clearly stating the number of papers identified in each database, the number of duplicates and the number of papers excluded along with the reasons for exclusion and finally the number of studies included.

RESPONSE:  This has been added accordingly.  

  1. Case reports – Please provide the exact follow-up period for each patient. The authors state that both patients were doing well at the 6-week follow-up visits. Since the surgery was performed in 2023, I would advise you to define the maximum follow-up time for each patient more precisely. This is important for further literature reviews or meta-analyses.

RESPONSE:  This has been added accordingly.  

  1. Abbreviations should be expanded when they are mentioned for the first time in the text (e.g. VA, COPD, BMI, etc.).

RESPONSE:  This has been modified accordingly.  

  1. Abbreviations that have been explained once in the text, such as preoperative progressive pneumoperitoneum (PPP), should not be explained again in the Discussion. In the Discussion, authors should only use abbreviations, otherwise the use of abbreviations makes no sense.

RESPONSE:  This has been modified accordingly.

  1. Table 1 – Each abbreviation used in Table 1 should be presented in a table legend. Some abbreviations have not been explained (e.g. BTA, LOS, CAD, DM, etc.). The legend should be placed below the table, not above it.

RESPONSE:  This has been done accordingly.

  1. The authors should add a few lines in the discussion of how botulinum toxin is helpful in treating these giant hernias and the benefits of this treatment compared to treatment without the use of botulinum toxin.

RESPONSE:  This has been done accordingly.

Reviewer 2 Report

Comments and Suggestions for Authors

I have read with interest the article by Dr. S Huerta et al entitled “Botulinum Toxin A as an Adjunct for the Repair Giant Inguinal Hernias: Case Reports and a Review of Literature” in which they describe their experience with the use of botulinum toxin with prehabilitation of 2 patients with giant inguinoescrotal hernia and reviewing the literature.

This is a rarely published indication of botulinum toxin, only 13 cases according to their review. Perhaps this is because the use of botulinum toxin for this indication is outside its technical specifications and, in my opinion, it is not the best proposal for its management. Regardless of the hernial size, the problem in these giant inguinal hernias is the volume of the hernial content and not the size of the hernial defect, so the use of progressive preoperative pneumoperitoneum would be more indicated with the idea of ​​avoiding abdominal hyperpressure than the toxin, since the hernial defect can usually be repaired without problems.

I am especially struck by the fact that throughout the article any mention is made of such important aspects as the Tanaka index or the guideline of the European Hernia Society (EHS) on inguinal-scrotal hernia, as well as not finding any imaging test in any of the cases and not seeing the keyword “inguinoscrotal hernia” in the literature review.

I believe the main limitation of the manuscript is the small number of patients, both from their own experience (2 cases) and from the literature review (13 cases).

In case number 1, a mesh with a size of 3x6 cm is used, totally contrary to current recommendations on the size of the mesh to be used (it does not provide information on case 2).

From the point of view of references, I think that the number of articles used (40) is excessive with a clear tendency towards self-reference (7 self-references).

Comments on the Quality of English Language

No comments.

Author Response

I have read with interest the article by Dr. S Huerta et al entitled “Botulinum Toxin A as an Adjunct for the Repair Giant Inguinal Hernias: Case Reports and a Review of Literature” in which they describe their experience with the use of botulinum toxin with prehabilitation of 2 patients with giant inguinoescrotal hernia and reviewing the literature.

RESPONSE:  We appreciate the candid comments from reviewer#2.  It is only with these comments that we can become better writers, and better practitioners.  Especially when dealing with a new technique that is outside of the typical conventional management in the management of a surgical issue. 

This is a rarely published indication of botulinum toxin, only 13 cases according to their review. Perhaps this is because the use of botulinum toxin for this indication is outside its technical specifications and, in my opinion, it is not the best proposal for its management.

RESPONSE:  We recognize this.   BTA has been extensively used for the management of complex ventral hernias and it is an accepted treatment today for that.  The use of BTA for ventral hernias started with a single case report or a couple of them.  Today, BTA for ventral hernias is widely accepted for the management of some complex ones.

BTA has not been used extensively for inguinal hernias.  It has not gained substantial traction in high-income countries (HICs).  This is because large inguinoscrotal hernias in HICs are not as common as ventral hernias.  Large inguinoscrotal hernias are more common in low- and middle- income countries (LMICs).  However, cost is a limitation for the use of BTA in LMICs. 

However, we wanted to publish our experience on this to let others know that this is feasible.   Further, we are not proposing this as an entire replacement for current accepted practices.  We are investigating this as a possible and feasible technique to be added to the armamentarium of modalities in the management of very large inguinoscrotal hernias.  The low side effect profile and its reproducibility should be appealing.  However, cost might be a major limitation, especially in LMICs. 

We have made all of this clearer in the manuscript and we have even added a segment that outlines the limitations of the study.   

Regardless of the hernial size, the problem in these giant inguinal hernias is the volume of the hernial content and not the size of the hernial defect, so the use of progressive preoperative pneumoperitoneum would be more indicated with the idea of ​​avoiding abdominal hyperpressure than the toxin, since the hernial defect can usually be repaired without problems.

RESPONSE:  I am afraid there is some misunderstanding on the utility of BTA for inguinal hernias.  BTA has nothing to do with addressing the hernia defect.  BTA utilization, in this setting, is purely to relax and augment the abdominal wall to allow for reduction of the hernia contents when there has been a loss of domain.  This exactly the same principle as preoperative pneumoperitoneum.   

I am especially struck by the fact that throughout the article any mention is made of such important aspects as the Tanaka index or the guideline of the European Hernia Society (EHS) on inguinal-scrotal hernia, as well as not finding any imaging test in any of the cases and not seeing the keyword “inguinoscrotal hernia” in the literature review.

RESPONSE:  I appreciate this comment.  As a prior participant of guidelines for other surgical issues, I am aware of the importance of guidelines for common surgical problems such as common and elective inguinal hernias.   However, it is the usual cases where we must step outside of guidelines.  It is also pivotal to recognize that a great deal of guideline recommendations DO NOT stem from LEVEL I evidence.   We are presenting in the following manuscript an unusual intervention.  I have undertaken this analysis not based on lack of experience or lack of not following current guidelines.  I assure the reviewer that over the past 20 years, as a practicing general surgeon, I have repaired over 2,000 hernias in complex veteran patients.  In fact, I was the first to implement a local anesthesia program in our hospital for the management of inguinal hernias.  Guidelines and current Level I evidence dictate the local anesthesia is superior to general and regional anesthesia for the repair of inguinal hernias.  However, a great number of surgeons do not adhere to these recommendations.

We have made this cleared in the manuscript and have added a segment that addresses the limitations of the current manuscript.    

I believe the main limitation of the manuscript is the small number of patients, both from their own experience (2 cases) and from the literature review (13 cases).

RESPONSE: It is precisely because there are not too many cases on this subject that we believe this manuscript and our experience is worthy of publication.  My colleagues have been using BTA for ventral hernias for several years.  Today there is no potential for publication with their experience using BTA for ventral hernias.  We hope to continue to generate traction for BTA in the management of inguinal hernias.    

In case number 1, a mesh with a size of 3x6 cm is used, totally contrary to current recommendations on the size of the mesh to be used (it does not provide information on case 2).

RESPONSE: I am sorry, but my experience dictates otherwise. 

I have found the less mesh leads to a decrease in the rate of inguinodynia. This has not affected my rate of recurrence.   At the risk of criticism for citing myself on this matter, I will do so to show the reviewer that my rate of inguinodynia and recurrence with the small mesh that I typically use for ALL hernias has been acceptable.  The hernia defect whether it is a defect of the trasnversalis fascia, or an indirect hernia sac has never been an issue repairing large inguinoscrotal hernias.

  1. Huerta S, Patel PM, Mokdad AA, Chang J. Predictors of inguinodynia, recurrence, and metachronous hernias after inguinal herniorrhaphy in veteran patients. Am J Surg. 2016 Sep;212(3):391-8. doi: 10.1016/j.amjsurg.2016.01.036. Epub 2016 May 6. PMID: 27324385.

  1. Huerta S. The best strategy for the management of inguinodynia is prevention. Hernia. 2023 Dec;27(6):1619-1620. doi: 10.1007/s10029-023-02778-z. Epub 2023 Mar 27. PMID: 36973466.

From the point of view of references, I think that the number of articles used (40) is excessive with a clear tendency towards self-reference (7 self-references).

RESPONSE:  First, I would like to underscore that this is not simply a case report, but a review of the literature.  As a simple case report, 40 citations might be excessive, but as a comprehensive review, we find that this number might not be as high as perceived. 

Secondly, we live in a world where just about anything can be found in the literature to support what we would like to believe.  There is a crisis in publishing today regarding just about all that we encounter.  I tell my residents and junior faculty to be careful with what they read because just about anything can be published.  A major predictor of how much I can believe that someone might be credible emanates from their experience.  The fact that I am self-referencing more than usual for this paper is an attempt to tell the reader that this is NOT a group that has fixed 10 hernias in the past few years.  Inguinal hernia surgery has been the focus of my practice for the past 20 years.  I have personally repaired over 2,000 complex hernias in veteran patients at our institution.  Thus, we were hoping that the references on prior hernia repairs would be a reflection of experience.  I did not want to give the readers the impression that I started a practice on inguinal hernias a few months prior to deciding to use BTA.  However, we have taken this comment seriously and have reduced the number of citations to accommodate the concerns of the reviewer.   

Round 2

Reviewer 2 Report

Comments and Suggestions for Authors

I think the manuscript has improved with the revision carried out. Thank you for accepting my comments.